# Sperm Functional Status: A Multiparametric Assessment of the Fertilizing Potential of Bovine Sperm

**DOI:** 10.3390/vetsci11120678

**Published:** 2024-12-23

**Authors:** Timea Sarah Odinius, Mathias Siuda, Matthias Lautner, Claus Leiding, Stefan Neuner, Heinrich Bollwein, Eleni Malama

**Affiliations:** 1Clinic of Reproductive Medicine, Department for Farm Animals, Vetsuisse Faculty, University of Zurich, Winterthurerstrasse 260, CH-8057 Zurich, Switzerland; timeasarah.odinius@uzh.ch (T.S.O.); msiuda@vetclinics.uzh.ch (M.S.); heinrich.bollwein@uzh.ch (H.B.); 2AgroVet-Strickhof, Eschikon 27, CH-8315 Lindau, Switzerland; 3Besamungsverein Neustadt a.d. Aisch e.V., Karl-Eibl-Strasse 17-27, 91413 Neustadt an der Aisch, Germany; lautner@bvn-online.de (M.L.); leiding@bvn-online.de (C.L.); neuner@bvn-online.de (S.N.)

**Keywords:** bull sperm, quality control, flow cytometry, fertility

## Abstract

Artificial insemination (AI) with cryopreserved sperm is the most widely used technique in bovine assisted reproduction. Semen production centers apply strict quality control measures before and after the cryopreservation of a sperm batch to ensure that the sperm cells have retained their functionality and are capable of fertilizing the oocyte and producing a well-developing embryo. Viability, namely the integrity of the cellular plasma membrane, is one of the routinely assessed features of frozen–thawed spermatozoa, as it is associated with the ability of sperm to interact with the egg. In addition to viability, we aimed to evaluate the importance of other sperm functional characteristics post-thaw, i.e., the integrity of the DNA and the ability of viable cells to maintain low intracellular Ca^2+^ levels and functional mitochondria, for the outcome of AI. Our findings suggested that cryopreserved sperm with acceptable viability may still suffer from DNA damage that can adversely affect the success of AI. The inclusion of traits reflecting the functional status of viable sperm did not improve our ability to predict the outcome of AI. In concluding, the incorporation of DNA integrity assessment in the quality control of cryopreserved bovine sperm may further improve the prediction of sperm fertility.

## 1. Introduction

Artificial insemination (AI) with cryopreserved bovine sperm remains the most widely used biotechnology technique in the assisted reproduction of dairy cattle worldwide. The outcome of AI is influenced by three main biotic components: the fertility status of the female, the fertilizing potential of inseminated sperm, and the interaction of the sperm and seminal components with the female genital tract and gametes. Previous studies have clearly shown the contribution of the male to the success of AI [1]. However, despite advancements in understanding sperm physiology and bull fertility management, many AI failures are still linked to male sub-fertility [1]. In cases of persistent suboptimal performance, sires may need to be removed from the semen production line. Additionally, a significant portion of commercially produced cryopreserved semen doses is rejected during post-thaw quality control [2], resulting in substantial financial losses and the loss of valuable genetic material.

In pre- or post-freeze quality control schemes, semen production centers examine various quantitative and qualitative characteristics to determine the suitability of sperm for further processing and AI [3]. These characteristics typically include the sperm concentration of the ejaculate, sperm morphology, motility, and plasma membrane integrity—commonly referred to as sperm viability. Sperm morphology is primarily assessed in neat semen, while motility and viability are evaluated both pre- and post-cryopreservation. The decision of whether to “discard” or “not to discard” an ejaculate or a cryopreserved sperm batch is often based on established cutoff values for motility or viability [3,4]. For AI in the field, frozen–thawed AI doses are expected to contain at least 30% progressively motile sperm [5], assessed using light microscopy [6] or computer-assisted sperm analysis (CASA) [7,8], and 40% viable sperm, defined as sperm with an intact plasma membrane and acrosome [3]. However, these two sperm quality traits are associated with varying fertility rates [9], and many consider that the fertilizing potential of a sperm batch cannot be adequately assessed based solely on sperm motility or viability [10,11]. As reported by Waterhouse et al. (2006), the presence of ≥7.5% DNA-damaged sperm in cryopreserved bovine AI doses led to an estimated 6% reduction in the odds of AI success [11]. Several studies have emphasized the importance of the sperm chromatin structure for producing viable and well-developing embryos [12,13,14,15,16]. DNA damage may not impair fertilization but can hinder the ability of the early embryo to complete the first cleavages [14] or form a blastocyst [13].

Recognizing the synergistic effect of different sperm functional features, researchers have explored holistic approaches for a more comprehensive characterization of bull fertility [9,10,17,18,19,20]. Given the multiparametric nature of male fertility, this appears to be a reasonable path to go down. As demonstrated by Sellem et al. (2015), a linear model combining flow cytometrically assessed data regarding sperm viability, oxidation, acrosome status, and mitochondrial function could explain up to 40% of the variation in bull fertility scores [18]. Bucher et al. (2019) further emphasized the link between sperm functional heterogeneity, in particular, the ability of viable sperm with functional mitochondria to regulate their cytocolic Ca^2+^ levels, and bull fertility after AI [10]. Nevertheless, before incorporating a new method into a sperm quality control scheme, two critical factors should be considered: first, the added value the new method will bring for bull fertility prognosis; second and equally important, the time and cost efficiency of the introduced method. Regarding the first point, many sperm functional traits are strongly correlated, providing overlapping information about the fertilizing competence of a sperm population [21]. Additionally, the importance of a sperm quality control method is evaluated in terms of its contribution to bull fertility prognostics and its applicability in routine practice. Therefore, elaborate techniques for sperm molecular profiling or multiparametric functional assessment may be considered technically demanding, time-consuming, or prohibitively expensive, and thus not suitable for routine sperm quality monitoring in semen production centers.

Notably, methods and equipment once considered “advanced” have nowadays gained ground in routine sperm quality control. Following the adoption of CASA platforms, several semen production centers have invested in flow cytometers equipped with standard blue lasers or more. Moving away from microscopic evaluation, flow cytometric assessment of sperm viability is increasingly used in the quality control of neat and cryopreserved bovine semen. Showing a good correlation with sperm motility [10,21], post-thaw viability collectively reflects the sperm cell’s ability to survive processing stress and support the extranuclear cellular changes needed for fertilization [22]. Moreover, flow cytometry provides a deep insight into various sperm functional properties beyond plasma membrane and acrosome integrity [23,24], such as the levels of intracellular Ca^2+^ [25,26], a critical signaling agent of sperm capacitation and acrosome reaction; mitochondrial function [18]; oxidative status [27,28]; and DNA integrity [29]. This raises the following question: could the flow cytometric assessment of multiple sperm functional traits improve the predictability of bull fertility compared to conventionally assessed traits such as viability?

In this context, it cannot be overlooked that a well-structured sperm quality monitoring program bears another crucial task in addition to the characterization of sperm fertilizing potential: the early identification of bull, management, or environmental factors that can affect the quality of the produced AI doses [30]. Remarkably, the effect of several management and environmental factors, such as the age of introduction of a young sire into semen production, bull nutrition, transient heat stress, and changes in the semen processing procedure or equipment, can be reflected by altered semen quality [31,32,33,34,35,36,37,38]. Furthermore, important decisions related to sperm processing are based on the results of sperm quality control, for example, decisions regarding the suitability of an ejaculate for conventional cryopreservation or sex sorting, and the adjustment of sperm concentration in AI doses produced by high-fertility sires [39]. Regarding bull age, it is known that quantitative and qualitative semen characteristics can vary with advancing age [31,40]. The widespread application of genomic selection has made it possible to identify future breeding sires at a very young age [41]. At the same time, it has drastically reduced the testing time available to evaluate young sires for their semen quality. Therefore, there is an increasing need to characterize or even predict the reproductive potential of peripubertal males. Thus, established sperm quality control schemes often need to be revisited to ensure that they are equally effective for bulls of different ages, or that they can efficiently detect management- and environment-related sources of sperm quality changes.

In this direction, we performed a retrospective fertility study using post-thaw sperm quality control and field fertility data collected over a 10-year period in a commercial semen production center, where flow cytometrically assessed sperm viability was used as a main criterion of post-thaw sperm quality. Our goal was to evaluate the incorporation of (a) sperm DNA integrity and (b) other sperm functional characteristics, i.e., intracellular Ca^2+^ levels and mitochondrial function, in sperm quality control to improve the predictability of fertility in AI sires.

First, we hypothesized that DNA integrity assessment may enhance the fertility predictive value of viability in frozen–thawed sperm. For this purpose, we analyzed a dataset of cryopreserved batches with simultaneous measurements of their sperm viability and DNA integrity (dataset A). Second, we tested the hypothesis that the co-evaluation of specific functional traits (esterase activity, intracellular Ca^2+^ levels, and mitochondrial function) together with sperm viability can improve the predictability of bull fertility in a second cohort of cryopreserved semen batches (dataset B).

## 2. Materials and Methods

### 2.1. Semen Collection and Sperm Processing

The sperm samples examined in our study were obtained through the regular semen collection schedule of a commercial semen production center in Neustadt an der Aisch, Germany. Ejaculate collections were scheduled on Mondays, Wednesdays, and Fridays. Sires were allowed to mount on a dummy cow or a bull, and semen was collected with a disposable tube connected to a pre-warmed (38 °C) sterile artificial vagina. Neat ejaculates were examined for their volume as well as sperm concentration, progressive motility, and morphology, using phase-contrast microscopy at 100× magnification. Ejaculates with a sperm concentration of ≥300 × 10^6^ sperm/mL, a progressive motility of ≥70%, and morphological abnormalities of <20% were further processed and cryopreserved. Semen was diluted with an egg yolk-supplemented Triladyl^®^ extender (Minitube, Tiefenbach, Germany) to a concentration of 60–80 × 10^6^ sperm cells/mL and loaded in 0.25 mL plastic straws (IMV Technologies, L’Aigle, France). Subsequently, semen collected on Monday or Wednesday was equilibrated for 24 h at 4 °C, while semen collected on Friday underwent an extended equilibration of 72 h at 4 °C prior to freezing [42]. Regardless of the equilibration time, freezing was carried out using the programmable freezing chamber MT Freezer 2.0 (Minitube, Tiefenbach, Germany). The temperature gradually decreased at rates of 1 °C/min from 4 °C to 2 °C, 10 °C/min to −10 °C, 20 °C/min to −100 °C, and 10 °C/min to −140 °C. Thereafter, sperm straws were submerged in liquid nitrogen (−196 °C) for storage until further evaluation.

In total, 10,427 cryopreserved sperm batches produced by 967 sires between 2012 and 2022 were included in this study (24 ± 52 batches per bull, ranging from one to 611 batches per bull throughout the period of 2012–2022) (Appendix A). Out of this pool, batches with complete records of their sperm viability and DNA integrity or sperm viability and functional status were considered for analysis in datasets A and B, respectively.

### 2.2. Sperm Quality Assessment

#### 2.2.1. Reagents and Samples’ Preparation

The chemicals necessary for preparing the Tris-NaCl-EDTA (TNE) buffer (0.01 M Tris, 0.15 M NaCl, 1 mM EDTA, pH 7.4), the acridine orange (AO) staining buffer (0.2 M Na_2_HPO_4_, 1 mM EDTA, 0.15 M NaCl, 0.1 M citric acid, pH 6.0), the acid detergent solution (0.15 M NaCl, 0.08 N HCl, 0.1% Triton-X 100, pH 1.2), as well as the fluorescent probes fluorescein isothiocyanate (FITC)-conjugated *Arachis hypogaea* lectin (PNA) and propidium iodide (PI) were purchased from Sigma-Aldrich Co. (Buchs, Switzerland). The fluorochromes CELLTRACE Calcein Violet AM, Fluo-4 AM, and 1,1′,3,3,3′,3′-gexamethylindodicarbocyanine iodide [MitoProbe DiIC_1_(5)] were obtained from Thermo Fisher Scientific Inc. (Waltham, MA, USA). The AO and the phycoerythrin-conjugated PNA (PE-PNA) were purchased from Polysciences Europe GmbH (Eppelheim, Germany) and GeneTex Inc. (Irvine, CA, USA), respectively. Working solutions were prepared for the following fluorescent probes: 6.0 mg AO/mL of staining buffer, 2.99 mM of PI, and 100 mg/mL of FITC-PNA.

From each cryopreserved batch, four straws were thawed in a water bath at 38 °C for 30 s and then pooled in a pre-warmed (38 °C) 1.5 mL laboratory tube. Samples were analyzed immediately after thawing.

#### 2.2.2. Dataset A

##### Plasma Membrane and Acrosome Integrity

Using the CytExpert Software for CytoFLEX version 2.1 (Beckman Coulter Inc., Nyon, Switzerland), the sperm plasma membrane and acrosome integrity were analyzed with the CytoFLEX^®^ Flow Cytometer V5-B5-R3. For this, cells were stained with the membrane-permeable dye PI and the acrosomal stain FITC-PNA, a FITC-conjugated lectin that binds to the outer acrosomal membrane. Unlike sperm with intact acrosomes, cells with defective acrosomes emit orange fluorescence. The flow cytometer, equipped with solid-state diode lasers operating with a power of 80 mW (violet), 50 mW (blue), and 50 mW (red), included five channels from the violet (405 nm) laser, five from the blue (488 nm) laser, and three from the red (638 nm) laser. The flow rate was established at 60 μL/min, with an analysis rate ranging from 500 to 1000 events/sec; 10,000 cells were analyzed for each sample. Five μL of semen were diluted in 241 μL of Tyrode’s solution and, thereafter, stained with 2.5 μL of FITC-PNA (working solution with 100 mg/mL of FITC-PNA) and 1.5 PI (working solution with 2.99 mM of PI). After a 15 min incubation at 38 °C in the dark, the samples were flow cytometrically assessed. Stained cells were excited with a 488 nm solid-state diode laser. The fluorescence emitted by PI and FITC-PNA was captured through 690/50 and 525/40 bandpass optical filters, respectively. Non-cellular events were excluded, and sperm events were selected based on side (SSC) vs. forward scatter (FSC) gating, as presented in Appendix A [28]. Then, plots of the FSC height vs. FSC area and the SSC height vs. SSC area were used for doublet discrimination and exclusion (Appendix A, respectively) [28]. Sperm that were PI- and FITC-PNA-negative were considered viable with an intact acrosome (PMAI), and were further used as input for data analysis.

##### Sperm Chromatin Structure Assay^TM^ (SCSA)

The susceptibility of sperm to acid-induced DNA fragmentation was evaluated through the SCSA, as previously described by Evenson and Jost (2000) [43]. For this, a Cytoflex^®^ flow cytometer (Beckman Coulter, Fullerton, CA, USA) was used, operated with the CytExpert Software for CytoFLEX version 2.1 (Beckman Coulter Inc., Nyon, Switzerland). In brief, 200 μL semen, previously diluted with TNE buffer to achieve a final concentration of 1 to 2 × 10^6^ sperm/mL, were treated with 400 μL of ice-cold acid detergent solution. After 30 s of mixing, 1.2 mL of AO working solution (6.0 mg AO/mL staining buffer) was added and the stained samples were evaluated through flow cytometry after exactly 3 min incubation on ice. Cells were excited using a 488 nm solid-state diode laser and the emitted green and red fluorescence signals were captured by means of a 525/40 and a 610/20 bandpass filter, respectively. A total of 10,000 cells were analyzed for each sample at a flow rate of 200 cells/sec. The analysis of flow cytometric data was performed using the FCS Express software (De Novo Software, Los Angeles, CA, USA). The percentage of cells with a high DNA fragmentation index (%DFI) was computed as previously described by Everson and Jost (2000) [43]. Technical details of the staining protocol, the configuration of the instrument, and the data processing approach used to perform the SCSA are described in Appendix A.

#### 2.2.3. Dataset B

##### Flow Cytometric Assessment of Sperm Functional Attributes

A five-color staining panel was employed for sperm functional analysis as previously described by Bucher et al. (2019) [10]. For this, we used a CytoFLEX^®^ Flow Cytometer V5-B5-R3 operated with the CytExpert Software for CytoFLEX version 2.1 (Beckman Coulter Inc., Nyon, Switzerland). The flow cytometer, equipped with three solid-state diode lasers of 405 nm (violet), 488 nm (blue), and 638 nm (red) operating with a power of 80 mW, 50 mW, and 50 mW, respectively, included five channels from the violet laser, five from the blue laser, and three from the red laser. The flow rate was set at 60 μL/min, with an analysis rate ranging from 500 to 1000 events/sec; 10,000 cells were analyzed for each sample. Dot plots of the SSC height vs. FSC area and SSC height vs. SSC area were used to exclude doublets [28].

For the simultaneous assessment of intracellular esterase activity, plasma membrane integrity, acrosomal status, intracellular Ca^2+^ levels, and mitochondrial membrane potential (MMP), sperm were simultaneously stained with calcein violet AM, PI, PE-PNA, Fluo-4 AM, and DilC_1_(5). Calcein violet AM served as an indicator for intracellular esterase activity, an alternative marker for cell viability. After excitation with the violet laser, cells with high esterase activity emitted violet fluorescence that was detected through a 450/45 bandpass filter. Following excitation with the blue laser, PI fluorescence was detected through a 690/50 bandpass filter. The fluorescent signal of PE-PNA was detected through a 585/42 bandpass filter after excitation with the blue laser. The intensity of green fluorescence emitted from Fluo-4 AM-stained cells corresponds to the concentration of free intracellular Ca^2+^. Following excitation with the blue laser, sperm with low or high intracellular Ca^2+^ levels emitted green fluorescence of low or high intensity, respectively. This fluorescence was captured through a 525/40 bandpass filter. Integrated into the fluorescence panel, the cationic dye DilC_1_(5) served to identify sperm sub-populations with different MMP. Upon exposure to the red laser, sperm with high MMP emitted an intense signal in the far-red region, whereas those with low MMP emitted light red fluorescence; red fluorescence was captured by a 660/20 bandpass filter.

To analyze each sperm sample using the five-color staining panel, the frozen–thawed sperm was diluted to a concentration of 1.2 × 10^6^ sperm/mL with Tyrode’s solution, reaching a final volume of 244.75 μL in a 250 μL reaction well of a 96-well plate. Prior to performing the assay, the fluorescent probes were mixed in a staining solution, consisting of 0.375 μL of calcein violet AM (1.21 μM), 1.5 μL of PI working solution (2.99 mM), 0.5 μL of PE-PNA (1 mg/mL), 2.5 μL of Fluo-4 AM (2 μM), and 0.375 μL of DilC_1_(5) (0.015 μM) per reaction well. Thus, 5.25 μL of the staining solution were added to each reaction well. Samples were flow cytometrically assessed after 15 min incubation at 38 °C in the dark.

In this study, we focused on the sperm subpopulation C_pos_PI_neg_PNA_neg_F_neg_M_pos_ that simultaneously exhibited the following features: (a) high esterase activity (C_pos_); (b) an intact plasma membrane, not stained with PI (PI_neg_); (c) an acrosome not stained with PE-PNA (PNA_neg_); (d) low intracellular Ca^2+^ levels (F_neg_); and (e) high MMP (M_pos_). Furthermore, we quantified the PI_neg_PNA_neg_ subpopulation that corresponds to the PMAI sperm as a measure of sperm viability. The percentage of C_pos_PI_neg_PNA_neg_F_neg_M_pos_ and PMAI sperm in the total cell population was quantified for each cryopreserved batch immediately after thawing. Technical details of the five-color staining protocol, including the fluorochromes used, the configuration of the instrument, and the gating strategy used to assess the functional status of sperm are described in Appendix A.

### 2.3. Fertility Data

The fertility outcome of each sperm batch was evaluated between day 60 and 90 after AI. For each batch, the non-return rate (NRR) was calculated as the percentage of females that did not return to estrous within 60–90 days after insemination. The NRR values were not corrected for any factors related to the season of AI, the AI technician, the female, or the farm. The batch-specific fertility data were provided by the Besamungsverein Neustadt a.d. Aisch e.V., Germany. In total, NRR records were available for 24,443 cryopreserved sperm batches produced over the period of 2012–2022 (Appendix A).

### 2.4. Statistical Analysis

The R language and environment for statistical computing version 4.2.2 [44] was used for data analysis. Descriptive statistics are presented in the form of the mean ± standard deviation (SD) for continuous variables and the number of observations (absolute and % frequency) for different levels of categorical variables. The analyzed semen samples were assigned into three groups based on the bull’s age at the time of semen collection: young, with an age of <24 months (average age of 19.85 ± 2.89 months); mature, with an age of ≥24 months and <84 months (average age of 61.54 ± 15.07 months); and old, with an age of ≥84 months (average age of 95.94 ± 12.73 months). The distribution of each sperm quality trait was checked for deviations from normality by visual inspection (histogram, quantile-quantile plots) and by performing the Shapiro–Wilk test. The Pearson’s correlation coefficient r was computed to describe the correlation between continuous variables at a 0.05 significance level. We used the Kruskal–Wallis rank sum test to assess differences in sperm traits and NRR values between age classes. The Dunn–Bonferroni posthoc multiple-comparison procedure was performed when the null hypothesis of the Kruskal–Wallis test (H0: the mean ranks of the groups are the same) was rejected at *p* < 0.05.

For inferential statistics, the following independent variables were used as input for data analysis: bull age (continuous, in months), calendar season (categorical; summer, autumn, winter vs. spring), weekday of semen production (categorical; Wednesday, Friday vs. Monday), PMAI (continuous, %), %DFI (continuous, %; for dataset A), and C_pos-_PI_neg_PNA_neg_F_neg_M_pos_ sperm (continuous, %; for dataset B). For each cryopreserved sperm batch, the values of the NRR (continuous, %), as well as the NRR class built on five-unit intervals (categorical), were used as response variables. As shown in the Results section (3.1.2. Fertility data), we only used the values of the above-mentioned independent and response variables from batches that achieved a minimum of 100 first services. Thus, dataset A included PMAI and DFI% assessments of 791 frozen–thawed batches (Appendix A); the PMAI and C_pos-_PI_neg_PNA_neg_F_neg_M_pos_ values of 733 batches were included in dataset B (Appendix A). Up to 75% of the examined batches in datasets A and B were obtained from bulls up to the age of 78 and 75 months, respectively; the maximum age of sires was 122 and 116 months at the time of sperm batch production in datasets A and B, respectively.

To explore the fertility predictive value of different sperm quality traits, we performed all-subsets regression, a model selection approach that consists of testing all possible combinations of the predictor variables [bull age, season, weekday of semen production, PMAI, and %DFI at 0 h (for dataset A), and bull age, season, weekday of semen production, PMAI, and C_pos_PI_neg_PNA_neg_F_neg_M_pos_ sperm (for dataset B)]. Then, we selected the best model according to the Mallows’ C_p_-statistic criterion. The regression models were built using the regsubsets function of the leaps statistical package for R [45]. Models with the lowest C_p_ value were trained using k-fold cross-validation (with k = 5) to compute the prediction error of each model [cross-validation error (CV)]; for this, the train function of the caret package for R was applied [46]. Notably, all-subsets regression suggested that PMAI was a redundant predictor when PMAI and %DFI were used to predict NRR values or the NRR class (C_p_ = 2.42 vs. C_p_ = 4.00 for the models including %DFI or both PMAI and %DFI, respectively). Regarding the analysis of dataset B, the regression model that included both PMAI and C_pos_PI_neg_PNA_neg_F_neg_M_pos_ as predictors of the continuous values of the NRR showed the lowest C_p_ value (C_p_ = 2.16 with CV error = 5.28). Including the age of the sire slightly decreased the prediction error of the model (C_p_ = 3.00 with CV error = 5.27). We observed the same when using the NRR class as a response variable.

The relation between the fertility outcome at batch level (continuous NRR values) with the sperm traits was assessed by fitting linear mixed-effects models with the toolset of the lme4 statistical package [47]. To test our first hypothesis, PMAI and %DFI were included as fixed predictors of batch-specific NRR values; PMAI and C_pos-_PI_neg_PNA_neg_F_neg_M_pos_ were used as fixed predictors to test our second hypothesis. In both cases, the effects of bull age (continuous), season, and weekday of semen collection were considered fixed effects. The effect of the bull was included as a random effect. Model coefficients were considered significant at a 0.05 significance level.

Conditional inference trees were implemented to predict the the NRR class of individual batches based on their post-thaw PMAI and %DFI values (dataset A) or PMAI and C_pos-_PI_neg_PNA_neg_F_neg_M_pos_ data (dataset B). The age class of the bull on the day of sperm batch production was also included as a predictor. For this purpose, the toolset of the statistical packages party and partykit for R was used [48]. A non-linear NRR diagram was generated to illustrate a tree-shape probability map of NRR classes. The selection of the input variable for splitting was based on the univariate *p* values and a split was implemented only for *p* < 0.05, in accordance with Bonferroni’s criterion.

## 3. Results

### 3.1. Descriptive Statistics

#### 3.1.1. Sperm Quality Traits

Measures of descriptive statistics for the PMAI, C_pos_PI_neg_PNA_neg_F_neg_M_pos,_ and %DFI values of datasets A and B, in relation to the age class of the sire, are demonstrated in Appendix A, respectively. Importantly, sperm traits in both datasets showed similar measures of central tendency (arithmetic mean) and dispersion (SD) with these recorded for the pool of 10,427 batches examined over the 10-year monitoring period (Appendix A); thus, batches included in datasets A and B were considered representative of the overall batch production. The values of PMAI in datasets A and B ranged well above the suggested threshold of 40% for cryopreserved bovine sperm (56.35 ± 9.87% and 54.21 ± 10.27% for datasets A and B, respectively) [3]. The values of PMAI were significantly improved in mature vs. young sires but showed a decline in older bulls (*p* < 0.01 for all pairwise comparisons; Appendix A for datasets A and B, respectively). Similarly, older bulls showed significantly lower C_pos_PI_neg_PNA_neg_F_neg_M_pos_ values when compared to young and mature sires (Appendix A). On the other hand, the overall %DFI scores of dataset A were kept at levels lower than 8% (4.53 ± 2.38%; Appendix A), a value suggested by Bucher et al. (2019) as an upper threshold for AI doses with good fertility [10].

The correlation coefficients, the sample size, and the adjusted *p* values for the pairwise correlations between the sperm traits recorded for the pool of 10,427 cryopreserved batches are demonstrated in Appendix A. As expected, %DFI was negatively related to PMAI (assessed through a dual PI/FITC-PNA staining or a five-color staining panel; r = −0.16, *p* < 0.001, and r = −0.28, *p* < 0.001, respectively) and C_pos_PI_neg_PNA_neg_F_neg_M_pos_ (r = −0.17, *p* < 0.001; Appendix A). A weak negative correlation between PMAI and %DFI was also observed in dataset A (r = −0.17, *p* < 0.001; Appendix A). The values of PMAI and C_pos_PI_neg_PNA_neg_F_neg_M_pos_ sperm were strongly correlated (r = 0.92, *p* < 0.001 and r = 0.93, *p* < 0.001 for the pool of 10,427 samples and the 733 samples of dataset B, respectively).

#### 3.1.2. Fertility Data

Descriptive statistics of the batch-specific fertility records (the number of first services, NRR, and the breed of sire) are presented in Appendix A, in relation to the age class of the bull on the day of semen collection. Based on the records of 24,433 cryopreserved batches produced from 1429 sires over a 10-year period, a mean NRR of 63.53 ± 22.50% was calculated; nonetheless, this estimate included batches with a diverse number of first services (ranging from 1 to 764 first services). One of our goals was to determine the minimum number of first services required to obtain a reliable estimate of the NRR for a specific batch. For this purpose, the SD of the NRR was computed for different levels of this minimum at 10-step intervals, i.e., a minimum of 10, 20, 30, etc. first services. The SD of the NRR for different minima of first services is demonstrated in Figure 1 and ranged from 3.29% to 8.27% (SD_mean_ of 4.83%) (Figure 1). Batches with less than 75 first AIs showed a considerably higher SD of their NRR estimate, i.e., more than one standard deviation away from the SD_mean_. On the contrary, when estimating the NRR for batches with 100 to 500 first AIs, the SD of the NRR remained relatively constant. This was not the case for batches with a minimum of 650 first AIs, which showed a rise in the NRR variation. At closer inspection, all batches with ≥650 first AIs were produced from mature Fleckvieh sires (older than 4 years) and scored an NRR of ≥ 69%, except for one sperm batch collected from a 16-month-old bull with a considerably lower NRR of 55%; apparently, this was the reason for the elevated variation of the NRR of batches with ≥650 first AIs. Therefore, we further used only batch-specific fertility data derived from at least 100 first services as input for our analysis, i.e., NRR records for 8,751 semen batches produced by 1,151 sires over the period of 2012–2022. Notably, the distribution of the NRR values for these batches was concentrated around the mean NRR with an SD of 5.58% (mean NRR 63.28 ± 5.58%) (Figure 2). As shown in Figure 2, only a small fraction of batches (with a minimum of 100 first services) had an NRR of <55% or >70%; these batches represented less than 7.5% of the 8,751 batches, respectively. The NRR values of batches examined in datasets A and B also showed a very similar distribution with mean NRRs of 62.02 ± 5.40% and 62.88 ± 5.33%, respectively (Appendix A, respectively). The variance of the NRR values in datasets A and B, as indicated by the SD measure, was higher for batches produced from young vs. mature and older sires (Appendix A).

### 3.2. Contribution of DNA Integrity Assessment to Fertility Prediction

#### 3.2.1. Linear Model

As shown in Table 1, the NRR scores increased with the advancing age of the bull (b = 0.03, *p* = 0.001). Batches produced during fall appeared to result in a higher NRR compared with spring batches (b = 1.58, *p* = 0.001; Table 1). On the other hand, the NRR of cryopreserved AI doses produced in summer or winter did not significantly differ from the NRR recorded for spring AI doses (*p* = 0.640 and *p* = 0.103, respectively; Table 1). Elevated %DFI values had an adverse effect on NRR values (b = −0.31, *p* = < 0.001; Table 1, Figure 3), but, as also indicated by preliminary all-subsets regression analysis, we did not detect any link between the NRR and PMAI (b = 0.01, *p* = 0.508; Table 1). Interestingly, sperm batches that were collected on Friday, and thus subjected to longer equilibration prior to freezing, showed significantly lower NRR than batches produced on Monday (b = −1.58, *p* = 0.001; Table 1). To capture any interrelation between the %DFI and progressing age, we also fitted a model that included the fixed effect of the interaction term *%DFI* × *age class* (young and old vs. mature bulls) on NRR values. The outcome of this model did not show any significant link between the NRR and the interaction term. Thus, the relation of the NRR with the %DFI was considered uniform for all age classes.

#### 3.2.2. Classification Decision Tree

We built non-parametric decision trees for the prediction of the NRR class, using PMAI, %DFI, and the age class of the bull as predictors, based on the analysis of 791 sperm batches with full PMAI, %DFI, and batch-specific fertility records after ≥100 first services (dataset A). This dataset included NRR records ranging from 34% to 76%, practically covering more than 95% of the fertility range observed in the sire population of the semen production center. The mean ± SD PMAI was 56.35 ± 9.87%, with 95% of the analyzed batches showing a PMAI of >37.4% (5th percentile 37.39% PMAI). Remarkably, the examined batches scored in a rather narrow range for sperm viability but showed more variable DNA integrity (coefficient of variation 17.5% and 52.5% for the PMAI and %DFI values in dataset A, respectively). A decision tree built with PMAI as the single predicting factor of the NRR class was inconclusive; hence, a tree with a depth and height of zero was created. Therefore, the PMAI score of a batch could not be used alone to predict the NRR class. Nevertheless, the addition of the %DFI significantly improved the decision tree model. Batches with a %DFI of >6.86% were predicted to score in the 55–60% NRR range with 49.3% probability (*p* < 0.001), while the probability of achieving an NRR higher than 60% was less (Figure 4). In contrast, semen batches with a %DFI of ≤6.28 had only a 23.1% probability of showing an NRR of 55–60% (*p* = 0.03; Figure 4) but were more likely to score an NRR of >60% (38.8% and 24.5% probability to achieve an NRR of 60–65% and 65–70%, respectively; Figure 4). Semen batches with 6.28% < %DFI ≤ 6.86% were also more likely to reach an NRR of 60–65% and 65–70%, with a probability of 35.1% and 32.4%, respectively (Figure 4). Interestingly, 7.46% of the examined batches with a PMAI of ≥40% failed to keep their post-thaw %DFI below the threshold of 6.86%. These batches had a mean NRR of 58.61 ± 5.00%, while batches with simultaneously PMAI ≥ 40% and %DFI ≤ 6.86% scored almost 4 units higher in the NRR scale (mean NRR 62.38 ± 5.15%).

### 3.3. Contribution of Sperm Functional Status to Fertility Prediction

#### 3.3.1. Linear Model

The analysis of 733 semen batches with simultaneous PMAI and C_pos-_PI_neg_PNA_neg_F_neg_M_pos_ records showed that none of the two sperm traits were significantly related to batch-specific NRR values (*p* = 0.204 and *p* = 0.571 for the b coefficients of PMAI and C_pos-_PI_neg_PNA_neg_F_neg_M_pos_, respectively; Table 2). On the contrary, the season and the weekday of semen collection appeared to be related to the NRR outcome. Particularly, semen batches produced during fall and winter were linked to higher NRR values (b = 1.18, *p* = 0.032 and b = 1.63, *p* = 0.002, respectively; Table 2). Batches collected on Friday were associated with significantly lower NRR scores (b = −1.83, *p* = 0.003; Table 2). The age of the sire did not play a significant role in NRR variation in dataset B (b = 0.01, *p* = 0.445; Table 2).

#### 3.3.2. Classification Tree

The decision trees built based on 733 combined records of PMAI and C_pos-_PI_neg_PNA_neg_F_neg_M_pos_ were of zero depth and height; thus, the combination of C_pos-_PI_neg_PNA_neg_F_neg_M_pos_ and PMAI could not reliably predict the probability of achieving one of the NRR classes.

## 4. Discussion

The analysis of our data revealed that the post-thaw DNA integrity and plasma membrane integrity of bovine sperm do not correlate with each other. Our findings are consistent with previous studies that also described a lack of correlation between these two traits [21,49,50]; however, conflicting findings have been also reported [11,51,52,53]. It is believed that DNA integrity is a non-compensable sperm characteristic [54] and reflects the structural stability of the nuclear genetic material, which is crucial for smooth embryonic development [55]. On the other hand, plasma membrane integrity is associated with the ability of sperm to survive cryopreservation stress and interact with the oocyte. Reduced sperm viability levels can be usually compensable by increasing the number of spermatozoa available for fertilization [21,49].

The weak link we observed between post-thaw viability and DNA integrity suggests that these two sperm features provide different but complementary information regarding the fertility of the examined AI doses. In the frame of a pre-freeze sperm quality control program, bull sperm are routinely checked and qualified for their membranes’ integrity or their motility before cryopreservation; this was also the case in our study. Thus, several batches with low sperm viability are already removed from the production line prior to cryopreservation and never reach post-thaw quality control. Nagata et al. (2019) showed that sperm populations selected for high motility and viability before freezing exhibited lower DNA damage post-thaw compared to the control [56]. In the same direction, Ruiz-Diaz et al. (2023) reported that the subpopulation of frozen–thawed bull sperm selected after thermotactic migration had lower DNA fragmentation and better fertility after intracytoplasmic sperm injection compared to the non-migrating cell subpopulation with lower motility [57]. Therefore, one can expect that DNA instability is not uniformly observed across sperm cells with diverse viability or motility. In our study, 94% of the examined batches showed ≥40% viable cells and, overall, a low variation of their sperm viability post-thaw (coefficient of variation 17.5%). Similarly, approximately 92% of the AI doses overall had a %DFI lower than the suggested threshold of 6.86%; however, we observed a comparatively higher variation of %DFI values post-thaw (coefficient of variation 52.5%). Thus, we cannot exclude that the rejection of neat semen samples on the viability criterion simultaneously acted as a positive selection on DNA integrity in our study. Nevertheless, as indicated by the different coefficients of variation computed for the two traits, a considerable fraction of the examined samples with acceptable post-thaw viability showed compromised DNA integrity (7.46% of 791 sperm batches) and lower NRR scorings. Interestingly, Bochenek et al. (2001) reported a higher variation of %DFI values in bulls with low vs. high reproductive performance [58]. Our observations undoubtedly support the importance of DNA integrity assessment, even in cryopreserved sperm that have successfully passed pre- and post-freeze screening for their viability.

A further point to consider when discussing the association between the viability and DNA integrity of cryopreserved sperm is the diverse impact that the freezing process may have on these two sperm features [27]. Cold shock is known to induce detrimental changes in the structure and function of the plasma membrane and acrosome, leading to either cellular death or the premature induction of unfavorable capacitation-like changes. Notably, Anzar et al. (2002) have shown that the freezing–thawing procedure can lead to the lysis and loss of a considerable fraction (up to 18%) of PI-positive sperm cells [49]. On the other hand, the chromatin structure appears to be more resilient against cryo-shock, with the percentage of cells with fragmented DNA increasing by up to 4 percental units in frozen–thawed vs. neat sperm samples [59,60]. Furthermore, there is good evidence that the susceptibility of the sperm DNA structure to cryo-injury or other stressful factors may not be evident immediately post-thaw but may only be detectable after subjecting sperm to short-term incubation stress [27,42]. In combination, viability and DNA integrity appear to be indispensable parts of sperm quality control after thawing, not necessarily showing identical variance in different cell subpopulations and thus providing complementary information about sperm cryo-resistance and fertility.

Through applying linear models, we showed that increased DNA fragmentation levels, assessed with the SCSA immediately after thawing, had an adverse effect on the AI outcome of cryopreserved semen batches. In contrast, we could not detect a similar effect of post-thaw sperm viability on the variance of NRR values. As discussed above, this could be partially attributed to the lower variation of the PMAI valuescompared to the %DFI values. Similarly, Waterhouse et al. (2006) could not establish a link between the batch-specific NRR and the viability of cryopreserved bovine sperm [11]. However, the authors observed that the odds for a successful AI outcome were considerably less for batches with high DNA damage, as assessed by the SCSA and the terminal deoxynucleotidyl transferase dUTP nick end labeling method [11].

We should not overlook the fact that a linear model-based assessment of the relationship between male fertility and sperm quality comes with certain limitations [61]. There is evidence that this relationship can follow a nonlinear pattern [62,63], i.e., changes in fertility may not be in direct proportion to changes in sperm quality parameters. This renders linear model-based analysis a less efficient approach for male fertility prognostics. A further point for discussion regarding sperm quality indices and NRR prediction is the establishment of threshold values for the former. Applying thresholds in linear regressors (e.g., sperm traits) for binary (e.g., high- vs. low-fertile) or multiclass classifications (e.g., multiple NRR classes) can be complicated. Therefore, in our study, we additionally employed non-parametric conditional inference tree classifiers to define the cutoff values of sperm traits that would increase the probability of occurrence of predefined NRR classes for the individual batches of a sire. Realistically, in many retrospective studies that use field fertility data, the assignment of sires or batches into well-distinguished high- vs. low-fertility groups can be challenging or not meaningful per se. This was also the case for the current study, where defining high- vs. low-performance batches would not be appropriate because of the strong concentration of the NRR values around the population’s mean. Instead, we considered that calculating the probabilities of batches to land in narrower NRR “zones” based on their sperm quality would be more relevant for fertility prognostics in practice. Furthermore, we focused on improving the accuracy of our prediction algorithms by including only batches with ≥100 first services, so that the NRR values would be representative of the fertilizing potential of the batch.

In agreement with the outcome of the linear regression models, our tree-structured classifier showed that %DFI but not PMAI values were relevant to the NRR scoring of each batch. Frozen–thawed batches with a %DFI of >6.86% had less than one-in-three chances to achieve an NRR higher than 60%. On the contrary, sperm samples with up to 6.86% %DFI were most likely to score higher than 60% NRR. Bucher et al. (2019) previously indicated that low fertility in bulls is associated with sperm DNA fragmentation values of 6.4% or higher, with sires producing batches with >8% %DFI having an extremely low likelihood of achieving a high NRR [10]. Puglisi et al. (2012) and Karoui et al. (2012) reported slightly higher %DFI cutoff values, ranging from 7% to 10%, for the percentage of DNA-compromised sperm in frozen–thawed bovine semen [64,65]. Through our analysis, we set a threshold of 6.86% for DNA-damaged sperm, which was comparable to those described in previous studies. Nevertheless, one should note that several factors can affect the DNA integrity of sperm collected in a semen production center, and these should be taken into consideration when suggesting cutoff values for sperm quality traits. For instance, station-dependent differences regarding the applied cryopreservation protocols and storage conditions [59,66,67], the bacterial load of neat or cryopreserved ejaculates [68], and the local microclimatic conditions [42,69] can be additional sources of variance for sperm quality and, thus, affect threshold establishment.

In the present study, one of the station-dependent factors that appeared to adversely affect batch-specific NRR values was the duration of sperm equilibration prior to freezing. This was 24 h for ejaculates collected on any weekday except for sperm collected on a Friday that was equilibrated over the weekend (for 72 h). Murphy et al. (2018) previously reported a decreasing trend in the calving rate, from 50.5% to 48.3%, after AI with 24 or 72 h equilibrated bovine sperm, respectively [35]. Recently, by analyzing sperm quality data from the same semen production center, we were able to demonstrate an adverse effect of prolonged equilibration on the DNA integrity of frozen–thawed sperm following 3 h of incubation, especially during the summer months [42]. Similarly, Fleisch et al. (2017) also described a subtle deterioration of mitochondrial function and DNA integrity in sperm equilibrated for 72 vs. 24 h, which was detectable after subjecting frozen–thawed sperm to 3 h of incubation stress [70]. On the other hand, it appears that sperm viability post-thaw is not adversely affected by extending the equilibration time from 24 to 72 h [35,70]. Taken together, our results suggest that prolonged equilibration may affect the quality of a sperm batch, presumably by interfering with the resistance of sperm chromatin structure to stressful factors, and this may be reflected in reduced fertility after AI.

The season of semen collection was also one of the fertility predictors tested in our study. Specifically, winter and autumn ejaculates were associated with higher NRR compared to spring and summer sperm batches. These findings are in line with previous research that has highlighted the consistent influence of season on semen quality and reproductive success in livestock [31,36,42,71]. Haugan et al. (2005) described that bovine AI doses produced in winter can achieve higher calving rates compared to those produced in summer. However, the effect of the season of semen collection may be evident in the calving rate but not necessarily in the NRR of a cryopreserved batch [72]. It is important to note that the findings of studies focusing on the effect of the season of semen collection on the AI outcome may not be directly comparable. Differences in husbandry conditions, the age and genetic background of the sires studied, local microclimatic conditions, and the methodology used to generate sperm quality and fertility data make the direct comparison of NRR seasonality patterns between studies difficult, if not impossible [37]. Several of these limitations can be overcome when assessing the fertilizing potential of cryopreserved semen batches after in vitro fertilization. Indeed, sperm collected and cryopreserved during summer in regions with temperate climates produce embryos with lower developmental potential compared to winter sperm [73,74]. In the current study, we detected improved NRR values for batches produced in autumn compared to spring. This was observed not only for dataset A (predictors: viability, DNA integrity) but also for dataset B (predictors: viability, functional status assessed through multicolor flow cytometry). Notably, in a recent study by Cinar et al. (20–24), we revealed a distinct seasonal pattern for the viability and DNA integrity of the cryopreserved sperm, with both traits showing a significant deterioration in the summer vs. winter batches [42]. The extent to which seasonal variations in different sperm traits (compensable and non-compensable) contribute to the seasonal pattern of the AI outcome would be an interesting topic to investigate in follow-up studies.

In addition to seasonality, we also demonstrated a linear relationship between bull age and the AI outcome, in particular, an improvement in NRR scores with increasing bull age. A wide range of age classes were represented in our study, with batches from sires of mature age (between 24 and 84 months) making up the bulk of the data, closely followed by samples from younger (up to 24 months) and older sires (over 84 months). Age-dependent changes in the functional status, especially the DNA integrity, of bull spermatozoa have been previously well documented [65,75,76,77]. These alterations with progressing age have gained exceptional importance in the last decade due to the widespread use of genomically selected peripubertal sires for commercial semen production [63]. Karoui et al. (2011, 2012) described the evolution of several sperm characteristics, including motility and DNA integrity, across a wide range of age classes and highlighted the non-linear changes of sperm characteristics over time. Ejaculate and semen characteristics improve significantly up to 18 months of age, then reach a plateau in sexually mature sires (2–5 years of age) and continue to decline in older bulls [65,76]. In addition to the significant improvement in batch-specific NRR values with increasing age, we also observed a decline in NRR variation across successive age classes, especially for batches from sires aged >18 months. As discussed above, bulls up to this age may still not have achieved a stable performance in terms of semen quality; this could be a reason for our observation. However, other non-biological factors, such as the detection and removal of sub-fertile males from the production line at an early age and the different marketing or distribution of AI doses from proven mature sires vs. young sires, may also interfere with age-dependent variations in the NRR values. In an attempt to capture any non-linear or, as suggested above, non-biological relationship between the age of a sire and the AI outcome of a batch, we also included age (as a continuous variable measured in months and as a categorical variable with predefined age classes) as a predictor in the conditional inference tree analysis. This could have potentially allowed us to set different thresholds for sperm characteristics for different age classes. However, age was not found to be a significant predictor for either dataset A or B.

In this work, one of our main objectives was to investigate the predictive value of a set of sperm functional attributes for the better characterization of the fertilizing potential of a cryopreserved batch. We were not able to reliably predict the NRR class of a batch by combining sperm viability and the percentage of viable sperm with functional mitochondria and low intracellular Ca^2+^ content as fertility predictors. The ability of cells to maintain low cytosolic Ca^2+^ concentrations and high MMP is reciprocally related to the status of their plasma membrane, with the viable sperm subpopulation showing better Ca^2+^ homeostasis and mitochondrial function [10]. Furthermore, there is good evidence that the presence of a sufficient number of cells with uncompromised mitochondria and efficient Ca^2+^ regulation in bovine semen is critical for the fertility outcome [10,20,25,78]. To minimize variation in the NRR, semen production centers apply strict criteria for post-thaw sperm motility and viability. Therefore, it is not surprising that we observed a narrow distribution of sperm viability and, consequently, of the percentage of viable sperm with low Ca^2+^ levels and functional mitochondria. The relatively low variance of the two traits and the strong correlation between them may partly explain the inconclusive results of the linear modelling and decision tree analysis using them as predictors.

In conclusion, our results showed that the examined functional parameters (esterase activity, plasma membrane and acrosome integrity, cytosolic Ca^2+^ content, and mitochondrial function) were strongly associated with the viability of cryopreserved sperm. Therefore, the functional characteristics assessed did not significantly improve the predictability of sperm fertility after AI. This was not the case for sperm DNA integrity, which made a significant contribution to the prediction of fertility, even in cryopreserved batches that had successfully passed the post-thaw screening for sperm viability. Nevertheless, sire- and station-dependent factors, such as bull age, the sperm processing protocol, and seasonality, may have an impact on the NRR variation and interfere with the fertility prediction algorithms.

## Figures and Tables

**Figure 1 vetsci-11-00678-f001:**
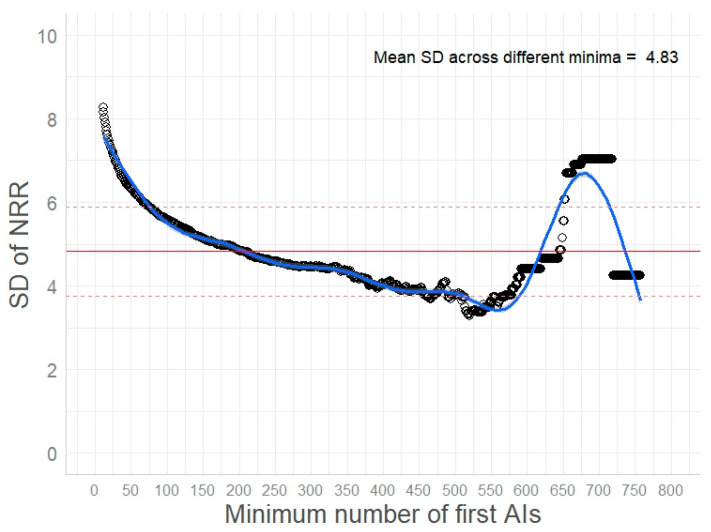
The standard deviation (SD, %) of the non-return rate (NRR) of cryopreserved batches 60 to 90 days after artificial insemination (AI), in relation to the minimum number of first AIs recorded for each batch. The mean SD of the NRR computed for different minima of the first services is represented by the horizontal solid red line; the dashed red lines reflect the SD values that lie one standard deviation away from the mean.

**Figure 2 vetsci-11-00678-f002:**
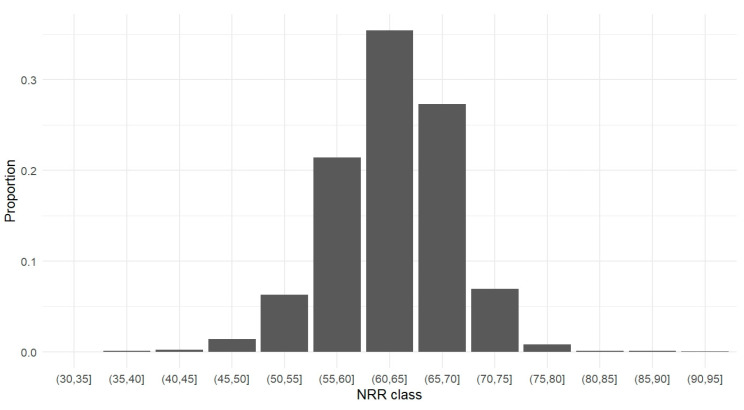
The proportion of cryopreserved batches (with a minimum of 100 first services) scoring different non-return rates (NRRs) 60 to 90 days after artificial insemination. The NRRs are demonstrated in the form of classes at 5-unit half-open intervals (they do not include their lower limit point).

**Figure 3 vetsci-11-00678-f003:**
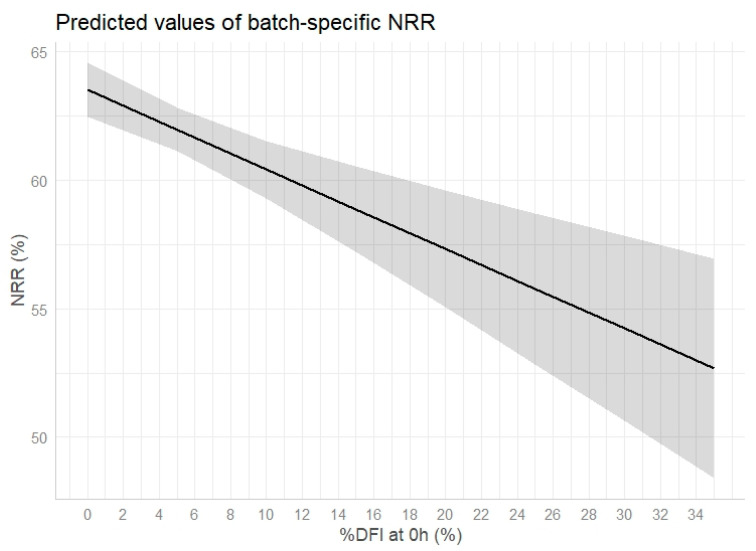
Predicted values of the batch-specific non-return rate (NRR, %) on day 60 to 90 post-insemination in relation to the percentage of sperm with a high DNA fragmentation index (%DFI, %) in the cryopreserved semen batch immediately after thawing (0 h). The plotted NRR values were predicted by fitting a linear mixed-effects model on the sperm quality data of 791 cryopreserved sperm samples produced from 236 sires and conditioned on the fixed effects of sperm viability, %DFI, bull age, season, and weekday of semen collection.

**Figure 4 vetsci-11-00678-f004:**
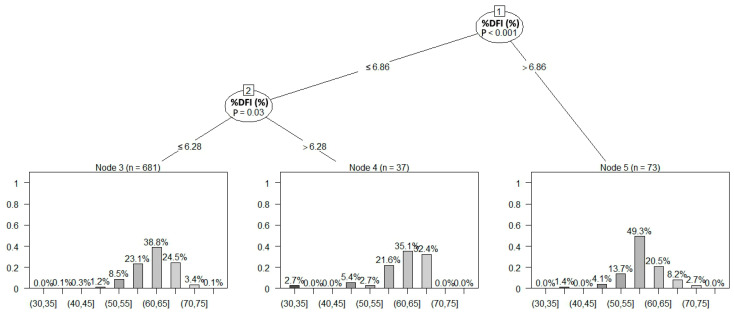
Decision tree illustrating the predictability of the non-return rate (NRR) class based on the percentage of sperm with a high DNA fragmentation index (%DFI). Classes of the NRR were built with 5-unit intervals (left-open and right-closed intervals) and included NRR scores between 30% and 80%. In each decision split node (nodes 1 and 2), the *p* value is presented for the selection of a significant predictor. The %DFI cutoff values are shown along the decision branch. The terminal nodes 3, 4, and 5 are in the form of bar plots showing the likelihood of a batch being assigned in a specific NRR class based on its %DFI score. The number of batches (n) landing in each terminal node is also presented.

**Table 1 vetsci-11-00678-t001:** Parameters [estimates of coefficient b with respective 95% confidence intervals (CI) and *p* values] of the linear mixed-effects model describing the relation between the non-return rate (NRR) 60 to 90 days post-insemination with cryopreserved semen and the age of the bull, the season, the weekday of semen collection, and the percentage of sperm with an intact plasma membrane (PMAI) or a high DNA fragmentation index (%DFI) in the semen dose immediately after thawing. The analysis was based on the sperm quality data and the batch-specific fertility data of 791 cryopreserved semen batches produced from 236 sires.

Effects on 60 to 90-Day NRR			
Predictors	Estimates	CI	p
(Intercept)	61.18	58.93–63.43	<0.001
Age (months)	0.03	0.01–0.05	0.001
%DFI	−0.31	−0.45–−0.17	<0.001
PMAI	0.01	−0.02–0.04	0.508
Spring	Reference		
Summer	0.19	−0.61–1.00	0.640
Fall	1.58	0.68–2.49	0.001
Winter	0.68	−0.14–1.49	0.103
Monday	Reference		
Wednesday	0.22	−0.63–1.07	0.608
Friday	−1.58	−2.55–−0.62	0.001
**Random Effects**			
Within-bull variance σ^2^		14.99	
Random intercept variance τ_00_ _bull_		8.53	
N_bull_		236	
Observations		791	
Marginal R^2^/Conditional R^2^		0.068/0.495	

**Table 2 vetsci-11-00678-t002:** Parameters [estimates of coefficient b with respective 95% confidence intervals (CI) and *p* value] of the linear mixed-effects model describing the relation between the non-return rate (NRR) 60 to 90 days after artificial insemination with cryopreserved semen and the age of the bull, the season, the weekday of semen collection, the percentage of sperm with an plasma membrane (PMAI), and the percentage of sperm with high esterase activity, an intact plasma and acrosomal membrane, low intracellular Ca^2+^ levels, and functional mitochondria (C_pos_PI_neg_PNA_neg_F_neg_M_pos_). The analysis was based on the sperm quality data and the batch-specific fertility data of 733 cryopreserved batches produced from 186 sires.

	Effects on 60 to 90-Day NRR		
Predictors	Estimates	CI	p
(Intercept)	58.88	56.48–61.28	<0.001
Age (months)	0.01	−0.02–0.04	0.445
%DFI	0.07	−0.04–0.17	0.204
C_pos_PI_neg_PNA_neg_F_neg_M_pos_	−0.03	−0.14–0.08	0.571
Spring	Reference		
Summer	−0.45	−1.61–0.71	0.445
Fall	1.18	0.10–2.25	0.032
Winter	1.63	0.62–2.64	0.002
Monday	Reference		
Wednesday	−0.02	−0.89–0.86	0.971
Friday	−1.83	−3.02–−0.63	0.003
**Random Effects**			
Within-bull variance σ^2^		18.93	
Random intercept variance τ_00__bull_		13.14	
N_bull_		186	
Observations		733	
Marginal R^2^/Conditional R^2^		0.038/0.432	

## Data Availability

Restrictions apply to the availability of these data. Data were obtained from the Besamungsverein Neustadt an der Aisch, Germany, and are available from the corresponding author with the permission of Besamungsverein Neustadt an der Aisch (https://www.bvn-online.de/de/index.html; accessed on 1 July 2024).

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
