# Peer review of "Sperm Functional Status: A Multiparametric Assessment of the Fertilizing Potential of Bovine Sperm"

_vetsci, 2024, doi:10.3390/vetsci11120678_

Round 1
Reviewer 1 Report
Comments and Suggestions for Authors
This manuscript describes an experiment to test the predictive potential of a multiparametric analysis of sperm quality data from DNA fragmentation, low Ca2+ levels, and active mitochondria on bull cryopreserved semen. The authors determine that DNA fragmentation obtained from SCSA is the most predictive variable.
This is an interesting and well-written manuscript. The authors correctly present the current situation in which AI centers have increased flow cytometry access, enabling new bull evaluation and selection strategies. Statistics are very detailed and based on FOSS, which is nice.
Another strong point is the vast number of analyses performed (however, be careful when interpreting P values when using a large n; see below).
I have some suggestions. While I believe this manuscript is worthy of publication with only minor corrections, I would like the authors to try some suggestions on sample analysis and data processing.
Please revise the text for (few) minor issues (e.g., L48 change "however" for "but"; L166 subscripts; L188 always add space between numbers and units).
Another suggestion for presentation is copying the supplementary tables to a text document and adding the captions. This would be more convenient for following the text, and additional explanations could be easily added if needed.
L156: Avoid using apostrophes in numbers. Instead, use a comma or a space for >4 digits.
P values in tables, including suppl.: indicate the exact P value, except for <0.001.
L186: "Unlike sperm with INTACT acrosomes"?
Methods:
Indicate the final sperm concentration for the cytometry analysis (not indicated by L192).
Detail the gating strategy for removing debris and do not refer to other studies.
Using a violet laser, add Hoechst 33342 for counterstaining; this is the most effective method of avoiding debris (calcein violet AM could be excluded or replaced by an alternative fluorochrome). This is the most problematic issue in this study and must be clarified. Please include representative cytograms for FSC/SSC, PNA/PI, and the calcein violet AM, PI, PE-PNA, Fluo4 AM, and DilC1(5) combination in the supplementary material.
Detail the compensation for the multiple fluorochrome combinations (suppl. mat.) since adjusting green, orange, and red fluorescence in the same line is problematic.
L237: PE-PNA
For the SCSA, were the samples evaluated at room temperature or refrigerated?
I suggest the authors use the opportunity to reanalyze the SCSA data (not for this study, but it could be possible with a batch—-automated-- analysis in no time) to obtain %HDS (a measure of chromatin compaction). This has been informative in many studies. It could also be considered in an additional study.
Indicate the final fluorochrome concentrations in all cases, not only the added volumes. This will make it easier for the reader to compare to other published protocols.
For Table S2, do you believe NRR can be considered a normal variable? I might have missed that (plots suggest it is). Otherwise, use the median and first and third quartiles.
The statistics are correct. Just a question: what is the basis for the bull age classification?
Finally, regarding the season, did the authors consider the offset from the onset of spermatogenesis to the collection/freezing of sperm? Samples collected in early autumn would come from summer spermatogenesis. Using the "spermatogenic" dates (50-ish days before collection) could be more adequate (I'm not sure if the authors did so).
L283: please indicate the P adjustment method.
L320: Just a tiny detail. As far as I know, the lme4 package does not provide P values for LMM (lmerTest does so, for instance), at least directly. Did you post-process the analysis results to obtain the P values?
Are L335-337 correct or from a draft?
In general, be wary of P values with a large number of observations. For instance, irrelevant correlations (PMAI vs. %DFI) show as highly significant without biological sense. However, the authors correctly comment on this low correlation in the discussion, so this is a minor issue.
Another comment on statistics. The fertility analysis, including %DFI and age, should include an interaction analysis (Table 1). Try testing the interaction with age as a discrete variable (Table S2). If significant, then split the data for young, mature, and old bulls. Considering that age and %DFI seem correlated, it might not be appropriate to mix them in a linear model simply.
L604-647: This is a very interesting discussion. However, it invites a deeper investigation of the relationship between %DFI and the day and season and an interaction analysis of the model (Table 1).
Reviewer 2 Report
Comments and Suggestions for Authors
Major points
1.Despite the mention of cryopreserved sperm in the importance of artificial insemination and the limitations of current sperm quality control methods, the background on why the special attention to DNA integrity and functional status of live sperm is not detailed enough.
2.The paper can further enhance the discussion of the specific mechanism of the effect of sperm quality on the artificial insemination success rate and the inadequacy of the existing research to enhance the pertinence and significance of the research.
3.The paper can specifically indicate how to apply the research results to the actual production.
4.When describing the data collection and processing method, the paper failed to specify the source of the data, the sample number, the processing process and the statistical analysis method.
5.The paper describes the process of collection, processing, staining and flow cytometry analysis of sperm samples, but some steps (such as the concentration of dye, incubation time, etc.) may lack detailed standardized instructions, which may affect the reproducibility of the experiment.
Minor points
1.the presentation of the results section is too lengthy or lacks structure, which is hard to read.
2.Line 693: Conclusion part should be summerized into one or two sentence rather than this paragraph.
3.Figure 4 are not clear which should be replaced by high quality picture.
4.The table should be corrected and delete the vertical line.
5.There are a large blank in page 10 which should be changed.
Comments on the Quality of English Languagethe english language of the whole paper should be revised carefully.
Reviewer 3 Report
Comments and Suggestions for Authors
Specific comments
Results
Page 7, lines 335-337, please delete these sentences.
Discussion
The author discussed on the differences of semen production day that production on Friday differed from production on the other day, and explained about the difference incubation times between 24 hours and 72 hours. However, I wonder if the laboratory boy/girl on these production days are the same person.
Reviewer 4 Report
Comments and Suggestions for Authors
Sperm viability is routinely assessed for the quality control of cryopreserved bovine sperm batches but is not usually conclusive about their fertilizing potential. This study investigated the fertility predictive value of bull sperm viability in combination with DNA integrity or the functional status of viable sperm. And they found that the incorporation of DNA integrity assessment can considerably improve sperm fertility prognostics, which guiding significance for production practice and for future research. However, there still some issues in this study:
1、There are many factors that affect AI efficiency, especially seasons. How did the author consider this in this study?
2、The incorporation of sperm DNA integrity is a key index for evaluating the
vitality and fertilization capacity of sperm. Did the author compare this index before and after cryopreservation?
3、in this study, Using the Flow Cytometer to analyze plasma membrane and acrosome integrity, suggesting to add relevant figures;
4、line 218, the headline "Flow cytometric sperm functional analysis", the word functional is not suitable, as the author only detected the esterase activity, plasma membrane integrity, acrosomal status, intracellular Ca2+ levels and mitochondrial membrane; These indicators are not functional;
5、The introduction part were too long, should be short;
6、the conclusion part is too complicated, suggest streamlining.
Comments on the Quality of English Language
The grammar of the entire paper seems to be fine, but some sentences are quite confusing. Some paragraphs are written too cumbersome。
